# Glass Foam from Flat Glass Waste Produced by the Microwave Irradiation Technique

**DOI:** 10.3390/mi13040550

**Published:** 2022-03-30

**Authors:** Adrian Ioana, Lucian Paunescu, Nicolae Constantin, Massimo Pollifroni, Dumitru Deonise, Florin Stefan Petcu

**Affiliations:** 1Engineering and Management of Metallic Materials Obtaining Department, Science and Engineering Materials Faculty, University Politehnica of Bucharest, 060332 Bucharest, Romania; nctin2014@yahoo.com (N.C.); miradeonise@yahoo.com (D.D.); petcu.stefan25@gmail.com (F.S.P.); 2Daily Sourcing & Research SRL Bucharest, 30609723 Bucharest, Romania; lucianpaunescu16@gmail.com; 3Management Department, University of Turin, 1106L31 Turin, Italy; massimo.pollifroni@unito.it

**Keywords:** materials, recycling, glass foam, flat glass waste, microwave heating

## Abstract

A glass foam with good thermal insulation characteristics (apparent density of 0.38 g/cm^3^, porosity of 81.9% and thermal conductivity of 0.089 W/m·K), high compressive strength (3.9 MPa) and a satisfactory microstructural homogeneity with pore size between 0.6–1.0 mm was obtained by sintering at 927 °C of flat glass waste, a glass waste usually not used in the manufacture of glass foam. The manufacturing recipe has been improved by the simultaneous use of two microwave susceptible foaming agents (SiC and Si_3_N_4_) and the addition of coal fly ash and an oxygen-supplying agent (MnO_2_). The originality of the work was the simultaneous use of the two foaming agents and also the application of the technique of predominantly direct microwave heating, compared to the conventional heating methods commonly used in the manufacture of glass foam. The remarkable energy efficiency of the microwave heating technique led to high average heating rates without affecting the structural homogeneity and very low specific energy consumption.

## 1. Introduction

Glass foam is currently a suitable solution for re-use as a newly created value material of glass waste available in large quantities mainly from post-consumer drinking bottles (or other container glass) and flat glass waste from demolition and renovation of buildings and from residual glass of the industrial production of flat glass or window glass. According to the literature [1], in 2018, the world amount of recycled glass waste was 27 million tonnes, and the recycling percentage of total glass waste represented 21% of this. The container glass waste had a recycling rate of 32%, while the flat glass waste had a recycling rate of only 11%. Most of the flat glass waste comes from the demolition and renovation of buildings, so it is a post-consumer waste. The current trend indicates an increase in the amount of this type of waste due to the massive replacement of old windows as well as glass walls and doors by modern double or triple-glazing for energy efficiency reasons. Recycling the flat glass cullet for industrial production of the new glass (25–50%) is not as extensive as the recycling process of container glass. The end-of-life building glass is usually recycled together with other building material waste and is recovered together with them or deposited in landfills. This waste can be used in the glass wool industry, the packaging glass industry, the glass bead industry (micro-balls of glass production), and as coarse aggregates for roads in public works [2,3]. In 2020 the European Commission adopted a plan whereby at least 70% of all construction and demolition waste will have to be re-used through recycling [4,5]. 

Generally, the properties of glass foam are remarkable, being a light weight material, a good thermal and acoustic insulator (low thermal conductivity and high porosity), with a high or at least acceptable compressive strength that is resistant to fire, humidity, freeze-thaw, corrosion, rodents, insects and bacterial aggression, with very high durability (above 50 years), and being non-deformable, non-toxic and chemically inert. Due to these properties, glass foam competes in terms of quality with the existing polymeric materials on the market, being used as light weight boards for building walls, floor tiles, paving, or as light aggregates for foundation fillings, underground thermal insulation for thermal pipes or storage tanks, road and railways construction, drainage, sports fields and many other applications [6,7].

In the last three decades, the industrial manufacture of glass foam has seen a significant increase, with sales of these products occurring almost worldwide. Several manufacturing companies (Misapor Switzerland, Pittsburgh Corning, Geocell Schaumglas, Glapor Werk Mitterteich) have consolidated their position and expanded their production capacity in several European countries (Belgium, Switzerland, Austria, Germany, Czech Republic, Nordic countries, etc.), the United States and China. The recipes for making glass foams of these manufacturers are focused on container glass waste and to a very small extent on flat glass waste, as the main raw materials. According to the literature [8], only Geocell Schaumglas and Glapor Werk Mitterteich are using a mixture of glass waste that also contains up to 10% flat glass waste. 

Among the methods applicable to foaming by expansion of glass waste, the most common is the method of incorporating a foaming agent into the starting powder mixture (there may even be two compatible agents), which at high temperature (generally, between 750–1150 °C) has the ability to release a gas. At the adequate temperature, the glass reaches the softening point and thus reduces its viscosity to a level where the gas spreads throughout the volume of the material, but does not leave it remaining blocked in the form of bubbles. By cooling, at the end of the heating process, the bubbles turn into pores, forming the typical structure of glass foam. According to [6], the most widely used foaming agents in industrial production are coal, black carbon, glycerol, silicon carbide and calcium carbonate. Many other pure inorganic or organic materials as well as by-products and wastes may be suitable as foaming agents, and currently researchers around the world are focused in this direction. 

Contamination of flat glass batches from building demolition (which may contain spacer bars from sealed units, silicon carbide discs, floor sweepings, etc.) must be kept to a very low level. The industrial manufacturers of glass foam are not as interested in this type of glass waste compared to container glass waste. Previous tests have shown that a significant increase of temperature and duration of the expansion process is possible due to some chemical components (including Fe_2_O_3_), which can influence the process parameters. Common methods of reducing these effects by using oxygen-supplying agents to improve the oxidation of the glass-based raw material can cause the macrostructural inhomogeneity of the foamed product [9]. The addition of the oxygen supplier greatly improved the foaming ability of Si_3_N_4_, used as a foaming agent for the soda-lime glass (the main commercial glass), leading to the decrease of the foaming temperature in the range 800–900 °C and the reduction of holding time at the foaming temperature at 7–30 min. The experimental results presented in [9] show that low values of the apparent density and compressive strength of glass foam can be obtained for higher values of the MnO_2_/Si_3_N_4_ molar ratio and enough high values of density (0.52 g/cm^3^) and compressive strength (4.4 MPa) corresponded to a much lower molar ratio. Heating the material to 850 °C for 30 min allowed for the obtaining of an acceptable uniformity of the sample microstructure, the maximum pore size being 1.6 mm. Heating to above 900 °C led to the impairment of the structural homogeneity determined by joining the neighboring cells. 

Experiments presented in [10] highlight the manufacture of glass foam from flat glass powder using combinations of foaming agents such as carbides and nitrides (SiC, Si_3_N_4_ and AlN) simultaneously with the addition of metal oxides with multiple valences (MnO_2_, Fe_2_O_3_ or CeO_2_), such as oxygen-supplying agents in addition to the oxygen in the oven atmosphere. The reduction by thermal decomposition of these oxides releases oxygen, which oxidizes the carbide or nitride generating a solid metal oxide and a gaseous oxide (NO_2_ or CO_2_), which expands the waste. The experiments concluded that the combined use of Si_3_N_4_ and AlN as foaming agents and the addition of MnO_2_ as an oxygen-supplying agent would be the optimal solution to promote both foaming and mechanical strength, obtaining glass foam with a homogeneous microstructure.

The experiments described in the literature [9,10] and presented above have been performed using conventional heating methods. Except for these works, between 2018–2021, research conducted by a team including the authors of the current paper from the Romanian company Daily Sourcing & Research SRL Bucharest was completed by the experimental production of glass foam from flat glass waste using several foaming agent types and additives (CaCO_3_ and SiC together with coal fly ash, respectively [11,12] and Si_3_N_4_ together with MnO_2_ [13]), the main difference compared to [9,10] being adopting an original predominantly direct microwave heating. In the work [11], a powder mixture from 98.6–98.8% flat glass waste and 1.2–1.4% CaCO_3_ wetted with 6.5–8% water addition was heated at 820–833 °C, resulting in a glass foam with an apparent density between 0.32–0.39 g/cm^3^, a compressive strength between 1.2–1.3 MPa and pore size between 0.6–2 mm. Also, the mixture based on flat glass waste including SiC between 3.5–3.6%, coal fly ash in the range 8.5–10.5% and water addition between 9–9.5% was heated at 980–995 °C, the product having apparent density between 0.32–0.36 g/cm^3^, compressive strength between 1.4–2.1 MPa, and a pore size between 0.9–3 mm. The paper [12] used SiC between 2.5–3.5%, coal fly ash in the range 9.5–10.5%, flat glass waste the rest and water addition of 9%. The sintering/foaming temperature was between 978–992 °C. The apparent density of the glass foam had a value between 0.25–0.31 g/cm^3^, while the compressive strength was between 1.27–1.35 MPa and the pore size was between 0.5–2.5 mm. The recent paper [13] included flat glass waste between 92.5–96%, Si_3_N_4_ of 2%, MnO_2_ between 2–5.5% and water addition of 14%. The mixture was heated at 815–860 °C. The glass foam has the following characteristics: apparent density between 0.36–0.57 g/cm^3^, compressive strength between 3.2–7.8 MPa, and pore size between 0.10–0.55 mm.

A brief analysis of the results obtained by the authors in the three papers above [11,12,13] showed that the application of the unconventional microwave heating technique of the raw material mixture favored obtaining glass foams with macro and microstructures with acceptable homogeneity regardless the manufacturing recipe. The use of coal fly ash contributed to improving the uniformity of the pore distribution but increased the temperature required for the sintering/expanding process to almost 1000 °C, caused by the Fe_2_O_3_ content (8.6%) of the ash due to the strong correlation between increasing the electrical conductivity of the glass and its ability to absorb microwaves. Also, the efficiency of microwave heating has been clearly improved by the high content of alkalis (Na_2_O, K_2_O) in the flat glass composition (above 13%) [14]. By comparison, the manufacturing recipe used in [13] with Si_3_N_4_ as a foaming agent together with the addition of MnO_2_ decreased the required process temperature below 860°C, but greatly reduced the pore size below 0.55 mm and excessively increased the compressive strength due to the denser character of the expanded product.

The microwave is not considered as an energy source, but an energy carrier, that in direct contact with a material with microwave susceptibility (called dielectric) undergoes the conversion of its power into heat. According to the literature [15,16], the initiation of microwave heating takes place in the core of the irradiated material and volumetrically propagates from inside to outside throughout its mass, which can be a very fast process. Also, this heating type has a selective feature, because it can be focused only against the material subjected to the heating, while the other massive components of the oven (vault, walls, hearth, etc.) are not heated.

Although discovered in the 19th century (between 1887 and 1889, based on a series of famous experiments, Heinrich Rudolf Hertz demonstrated the existence of electromagnetic waves predicted by Maxwell’s Theory, definitively confirming the theory of action by contiguity), electromagnetic waves including microwaves began to be used in telecommunications and radars in the 1930s and insignificantly in the heating processes of solids. Practically, microwave heating processes have meant drying or low temperature heating. According to the literature [17], at the beginning of this millennium it was known that several materials (ceramics, metals, organics, polymers, glasses, composites, etc.) are suitable for efficient microwave heating. However, the industrial application of this unconventional heating technique is delayed, with only concerns at small-scale of research teams being known, as is also the case of the research presented below in this paper. 

The aim of the current paper is the experimental small-scale manufacture of flat glass waste foam through technological changes in the current manufacturing process and the use of advanced techniques of predominantly direct microwave heating, as well as the combined use of SiC and Si_3_N_4_ as foaming agents (unlike the association between Si_3_N_4_ and AlN tested in [10]) together with MnO_2_ as an oxygen-supplying agent. The combination of silicon carbide and nitride, adopted for their high microwave susceptibility, has not been tried, much less in microwave heating conditions.

## 2. Materials and Methods

### 2.1. Materials

Analyzing the manufacturing recipes of glass foam from flat glass waste and the influence of different materials used in previous experiments as raw material or additives on the sintering/foaming process and the characteristics of the expanded product, the authors adopted the solution of the combined use of two foaming agents (SiC and Si_3_N_4_), coal fly ash as the raw material together with the predominant raw material (flat glass waste) as well as MnO_2_ as an oxygen-supplying agent. The solution of adopting two foaming agents from the carbide and nitride group is due to the results of the experiment presented in [10], but their choice was different from [10], being preferred for SiC and Si_3_N_4_, highly microwave susceptible materials. Coal fly ash was selected due to its favorable role in the foaming process of glass waste, helping to form a homogeneous microstructure, despite the increase in the required process temperature. This disadvantage was compensated by the choice of the oxygen-supplying agent (MnO_2_), which ensures the required gas released in the thermally softened mass of the glass and reduces the temperature of the expansion process.

The oxide composition of the raw materials (flat glass waste and coal fly ash) is presented in Table 1.

In principle, Fe_2_O_3_ in the composition of glass and coal fly ash is an inherent contaminant which influences the sintering/foaming process by increasing its temperature and disturbing the microstructural homogeneity of the glass foam [15]. Adopting the predominantly direct microwave heating technique, this disadvantage becomes advantageous because Fe_2_O_3_ is a microwave susceptible material especially at room temperature, compensating for the low susceptibility of SiO_2_ and Al_2_O_3_ at low temperatures (whose electrical conductivity increases with increasing the temperature, having optimal microwave absorption characteristics only after 500 °C [18,19]. Thus, the heating process begins with high efficiency at ambient temperature, although SiO_2_ and Al_2_O_3_ in the raw material are transparent microwave materials at this temperature.

Four experimental variants for the production of glass foam were adopted, varying the composition of recipes according to Table 2. 

According to Table 2, two foaming agents (SiC and Si_3_N_4_) were simultaneously used, both being recognized as highly efficient in glass foaming processes and, additionally, both being materials with high microwave absorption capacity, therefore able to provide heat inside the system by converting the wave power into heat. Keeping the Si_3_N_4_ ratio constant and slightly varying the SiC proportion, the SiC/Si_3_N_4_ weight ratio increased slightly from 1.33 (variant 1) to 2.33 (variant 4). The MnO_2_/Si_3_N_4_ weight ratio increased significantly from 1.33 to 3.33 ensuring conditions for a high foaming. The coal fly ash and SiC proportions slightly increased, but the ash/SiC weight ratio decreased from 3 (variant 1) to 2.57 (variant 4). Under these conditions, it is predicted to increase the foaming temperature value and the pore size as well as to decrease the apparent density and compressive strength.

Special attention was paid to the quality of the raw materials and additives used in this experiment. The flat glass waste was selected from a clear window glass waste, which was cleaned by washing, broken by manual methods and ground several times in an electric laboratory grinding device up to a granulation below 65 μm selected by sieving. 

The coal fly ash was purchased from a Romanian thermal power plant initially with a grain size below 250 μm. The grinding performed in the laboratory device led to the reduction of the ash granulation below 80 μm.

The silicon carbide (SiC) previously purchased from the market had a fine grain size below 10 μm, while the silicon nitride (Si_3_N_4_) also purchased from the market was a very fine powder (below 1 μm).

The manganese oxide (MnO_2_) used in this experiment was a very fine powder purchased from the market. 

The raw material and additives mixture was prepared separately for each adopted experimental variant by manual homogenization in a cylindrical vessel rotated in slightly inclined position, followed by wetting the mixture with water by adding the constant dosage of 14 wt. %. Immediately after wetting, the material was mixed with a metal rod and loaded by manual pressing into a cylindrical metal mold with removable wall with the inner diameter of 80 mm and then removed for free loading in the oven.

### 2.2. Methods

As mentioned above, the most common method of foaming glass waste is to release a gas in the thermally softened mass of the waste powder due to chemical reactions (decomposition, oxidation or other types) of the foaming agents incorporated in the waste mass. The two foaming agents adopted by the authors (SiC and Si_3_N_4_) are usually prone to oxidation reactions. Oxygen is usually taken from the oxidizing atmosphere of the oven or the free space between the fine particles of the mixture. According to [6], the oxidizing reactions of SiC that have optimal conditions for development at high temperatures (900–1100 °C), in thermodynamic terms, are:SiC + 2O_2_ = SiO_2_ + CO_2_
(1)
SiC + O_2_ = SiO_2_ + C (2)

SiO_2_ (solid) that resulted from reactions is taken over by the molten glass, while CO_2_ is the foaming gas that forms bubbles. According to [20], a reaction between C and CO_2_ may occur with the release of CO (gas) which also participates in foaming, forming bubbles in the molten mass of the glass.
C + CO_2_ = 2CO(3)

The addition of MnO_2_ as an oxygen-supplying agent is based on the successive reduction reactions by thermal decomposition to Mn_2_O_3_ (4), with a maximum at 590 °C and then to Mn_3_O_4_ (5) with a maximum at 800 °C [21].
4MnO_2_ = 2Mn_2_O_3_ + O_2_
(4)
6Mn_2_O_3_ = 4Mn_3_O_4_ + O_2_
(5)

The oxygen released by the two reduction reactions (4) and (5) oxidizes Si_3_N_4_ [13] according to reaction (6).
Si_3_N_4_ + 7O_2_ = 3SiO_2_ + 4NO_2_
(6)

According to one author [22], the Si_3_N_4_ oxidizing occurs according to reactions (7) and (8).
Si_3_N_4_ + 3O_2_ = 3SiO_2_ + 2N_2_
(7)
Si_3_N_4_ + 5O_2_ = 3SiO_2_ + 4NO (8)

Furthermore, reaction (9) is also possible.
2NO = N_2_ + O_2_
(9)

Therefore, in both variants, the solid product included in the composition of the molten glass is SiO_2_ and the gaseous products that participate in the foaming process are N_2_, O_2_ and/or NO_2_. 

The experimental manufacture of flat glass waste foam was carried out on a domestic microwave oven adapted for high temperature operation with a single magnetron whose waveguide is placed in one of the side walls of the oven. The upper metal wall of the furnace was provided with a central hole of 30 mm, whose axis communicates with the viewing axis of a radiation pyrometer mounted above the oven on a metal support at about 400 mm. The walls, hearth and flat vault of the oven have not been additionally thermally protected, as heat transfer takes place from the inside of the irradiated material to the outside, i.e., in the opposite direction to conventional heating methods. The pressed material containing the mixture of raw material and additives was freely deposited on a metal plate supported 15–20 mm above the thick thermal bed of ceramic fiber mattresses at the base of the oven. The material was protected from the overly intense radiation of the microwave field by a ceramic tube made of a mixture of SiC and Si_3_N_4_ and a strong microwave absorber, placed on the thermal insulation bed and covered with a lid of the same material with a central hole of 30 mm. The thermal protection of the heated material and the ceramic tube was done by covering the outer surface of the tube and the lid with thick ceramic fiber mattresses (resistant to 1200 °C). The 30 mm holes on the same vertical axis allowed the control temperature of the heated material during the sintering/foaming process by viewing with the pyrometer. The indication of the end of the foaming process is given by stopping the tendency of increasing the temperature on the upper surface of the material and the beginning of the decrease in its value. In 5–10 s, it is necessary to interrupt the power supply of the magnetron.

The overall image of the experimental microwave equipment described above and its constructive scheme are shown in Figure 1.

### 2.3. Characterization of the Glass Foam Samples

The methods of glass foam samples characterization were as usual. The apparent density was measured by the gravimetric method [23], and the porosity was determined comparing the values of apparent density and the compact material density [24]. The guarded-comparative-longitudinal heat flow method (ASTM E1225-04) was used for determining the thermal conductivity value. Using a TA.XTplus Texture analyzer, the compressive strength was measured. The water absorption was identified by the water immersion method (ASTM D570). The microstructural appearance of the samples was determined using a Smartphone Digital Microscope ASONA 100X Zoom type and the crystallographic structure was identified through X-ray diffraction (XRD) (EN 13925-2: 2003).

## 3. Results

The dry raw material amount was kept constant at 470 g and, implicitly, the wet amount had a constant value in all experimental variants (535.8 g). The main functional parameters of the experimental manufacturing process are presented in Table 3. 

According to the data in Table 3, the temperature of the sintering/expanding process increased from 920 °C in variant 1 to 942 °C in variant 4. The increase is explained by the presence of coal fly ash in the starting mixture. On the other hand, the use of the oxygen-supplying agent (MnO_2_) as a foaming intensifier contributed to the reduction of temperature values in all variants by about 30–40 °C. The process time, increasing from variant 1 to variant 4, had values lower compared to a foaming process using only coal fly ash and SiC, being in the range of 34–41 min. The heating rate was high (22.5–26.5 °C/min), generally well above the recommended values (around 10 °C/min [6]), without affecting the structural homogeneity. The volume growing of expanded products was significantly higher in the case of variant 4 and lower in the case of variant 1. Due to the excellent energy efficiency of the predominantly direct microwave heating method, the specific energy consumption had very low values, especially in the case of variants 1 and 2 (0.77 and 0.81 kWh/kg, respectively).

Images of appearance of the four samples obtained by the thermal sintering/expanding process according to the data in Table 3 are shown in Figure 2.

The glass foam samples were investigated by the characterization methods mentioned above, the main physical, thermal, mechanical and microstructural features being determined and presented in Table 4.

Experiments performed with the simultaneous participation of the two foaming agents (SiC and Si_3_N_4_), under conditions which included in the starting mixture both coal fly ash and SiC in glass foaming processes and MnO_2_ used together with Si_3_N_4_ in recent experiments [9,13], led to obtaining the appropriate expanded products in terms of quality. The apparent density did not exceed 0.49 g/cm^3^ in variant 1 with the lowest proportions of coal fly ash and MnO_2_, which would favor the glass foaming. The compressive strength reached the maximum value of 4.3 MPa, the thermal conductivity being in this case 0.110 W/m·K, i.e., a value low enough to ensure the thermal insulation property of the material. The products made in variants two and three had the apparent density in the range 0.32–0.38 g/cm^3^, porosity between 81.9–84.8%, thermal conductivity between 0.080–0.089 W/m·K, compressive strength between 3.5–3.9 MPa, and the pore size in the range of 0.6–1.1 mm, which is ideal for their use as thermal insulation materials in the building sector. The temperature of the foaming process in the two variants had quite high values due to the increasing proportion of coal fly ash. The disadvantage of the sample manufactured in variant 4 was the high temperature (942 °C) of the process and its longest duration. Also, the specific energy consumption reached the highest value (0.92 kWh/kg) compared to the other tested variants.

The comparative analysis of the microstructural configuration of samples manufactured in the four experimental variants was performed examining the pictures in Figure 3.

According to the pictures in Figure 3, generally the microstructural homogeneity characterizes all samples. The pore size was identified, having the values noted in Table 4. 

Considering the above observations, the choice of the optimal variant for the manufacture of glass foam from flat glass waste was oriented toward variants 2 and 3. Taking into account the lower specific energy consumption of variant 2 (0.81 kWh/kg), the authors’ option was variant 2.

The XRD analysis was performed only for the sample corresponding to this variant. The main crystalline phase identified after the thermal treatment at 927 °C was wollastonite-2M (CaSiO_3_) and traces of SiC and Si_3_N_4_ (Figure 4).

The formation of the crystalline phase of wollastonite in glass foam by using SiC as a foaming agent decreases the porosity and increases the mechanical strength of the foamed product. These effects influence to a small extent its physical and mechanical characteristics [25].

## 4. Discussion

The result of the research presented in the paper is the small-scale manufacture of glass foam from flat glass waste having the physical, thermal, mechanical and microstructural characteristics initially proposed. The product considered optimal (variant 2) had good thermal insulation properties (apparent density of 0.38 g/cm^3^, porosity of 81.9% and thermal conductivity of 0.089 W/m·K), a high compressive strength of 3.9 MPa, and water absorption of 3.3 vol. %, being suitable for use as thermal insulation material in the building sector competing with traditional products on the market. Also, the problem of the structural homogeneity of foamed materials from flat glass waste was solved, the expanded product having pores with dimensions in the range of 0.6–1 mm, which was satisfactorily distributed throughout the product volume.

The optimal manufacturing recipe included the simultaneous use of two foaming agents from the group of carbides (2.5% SiC) and nitrides (1.5% Si_3_N_4_). A combination of Si_3_N_4_ and AlN has recently been used with good results according to [10], but the solution adopted by the authors took into account that SiC and Si_3_N_4_ are highly microwave absorbent materials. The starting powder mixture also included a by-product (7% coal fly ash) commonly used with SiC and an oxygen-supplying agent (3% MnO_2_) recently used with Si_3_N_4_ to promote foaming. The optimum sintering/foaming temperature was 927 °C.

The remarkable energy efficiency of the predominantly direct microwave heating process had a major effect in terms of energy. The specific energy consumption was very low (0.81 kWh/kg), generally significantly lower than foaming processes by conventional heating techniques. The heating rate used for variant 2 was 25.2 °C/min, much higher than the rate recommended in the literature (around 10 °C/min), without affecting the microstructural homogeneity of the sample. The microwave heat treatment of glass waste for glass foam production has proven to be very energy efficient. In the perspective of future experiments, the authors intend to expand the processing capacity of the microwave oven to reach an industrial scale. 

## 5. Conclusions

The aim of the paper was to test some changes in the manufacturing recipe of glass foam on a small scale from flat glass waste and to apply an original technique of predominantly direct microwave heating. Also, the simultaneous use of SiC and Si_3_N_4_ as foaming agents, adopted for their microwave susceptibility, was an original element of the work. According to the literature, the flat glass waste is not normally used for the production of glass foam because it generates an inhomogeneous structure of the expanded product. On the other hand, the application of the microwave heating technique is not commonly used either industrially or in small-scale experiments for the manufacture of glass foam. The experimental results showed that by simultaneously combining two foaming agents (SiC and Si_3_N_4_), the use of an oxygen-supplying agent (MnO_2_) and coal fly ash to intensify the foaming as well as by applying the microwave heating method, obtaining products with good thermal insulation properties, high compressive strength, satisfactory microstructural homogeneity and very low specific energy consumption was possible.

## Figures and Tables

**Figure 1 micromachines-13-00550-f001:**
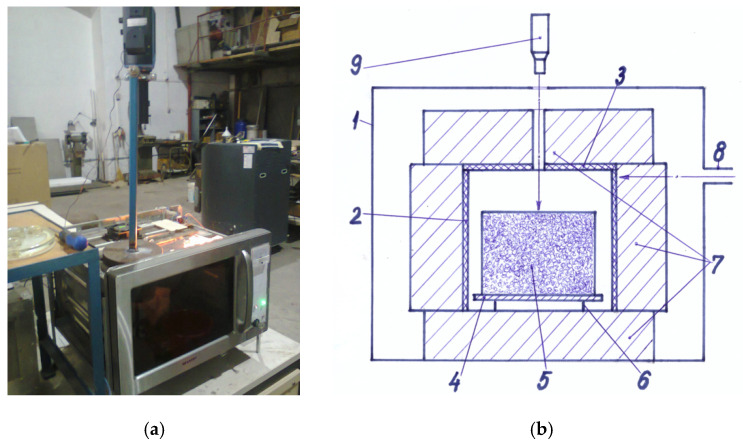
Experimental microwave equipment: (**a**) Overall image of the experimental equipment; (**b**) Constructive scheme of the experimental equipment: 1—0.8 kW-microwave oven; 2—ceramic tube; 3—ceramic lid; 4—metal plate; 5—pressed powder mixture; 6—metal support; 7—thermal insulation; 8—waveguide; 9—pyrometer (Reproduced by permission of Daily Sourcing and Research SRL).

**Figure 2 micromachines-13-00550-f002:**
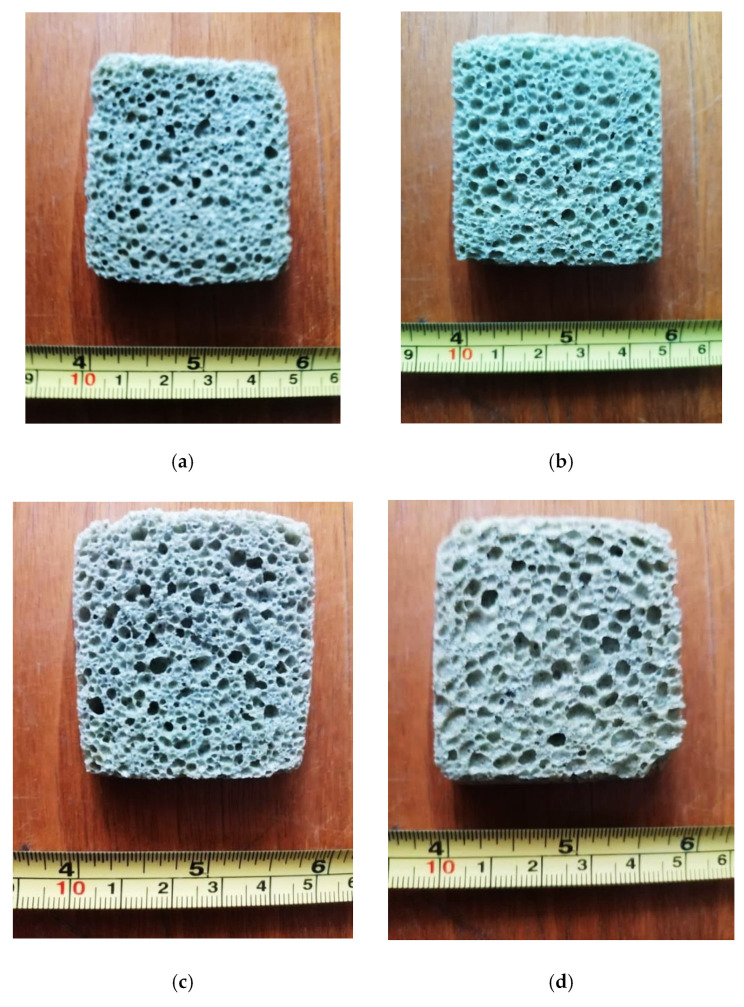
Appearance of the glass foam samples (cross section) corresponding to: variant 1 (**a**); variant 2 (**b**); variant 3 (**c**) and variant 4 (**d**).

**Figure 3 micromachines-13-00550-f003:**
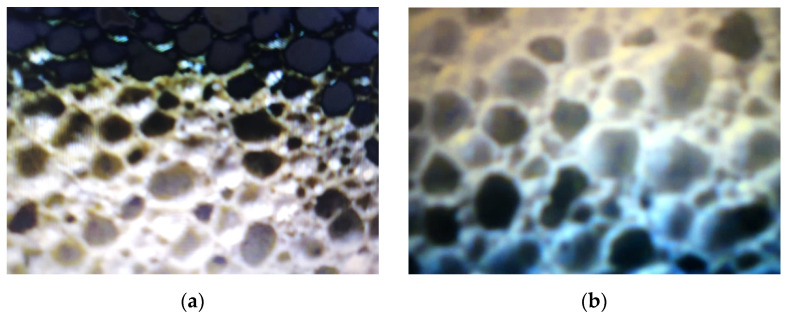
Microstructural configuration of the glass foam samples corresponding to: variant 1 (**a**); variant 2 (**b**); variant 3 (**c**) and variant 4 (**d**).

**Figure 4 micromachines-13-00550-f004:**
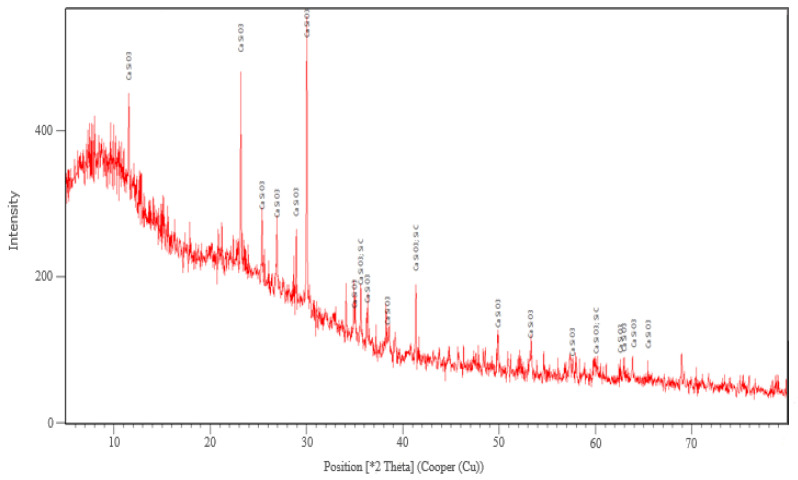
The XRD analysis corresponding to sample made in variant 2 heat treated at 927 °C.

**Table 1 micromachines-13-00550-t001:** Oxide composition of raw material.

Raw Material	SiO_2_	Al_2_O_3_	MgO	CaO	Na_2_O	K_2_O	Fe_2_O_3_
Flat glass waste (wt. %)	70–73	Max. 1.5	3.5–4.5	8–9.7	13.4–14.6		Max. 0.20
Coal fly ash (wt. %)	46.5	23.7	3.2	7.9	6.0	4.1	8.6

**Table 2 micromachines-13-00550-t002:** Experimental variants composition.

Variant	Flat Glass Waste wt. %	Coal Fly Ash wt. %	SiC wt. %	Si_3_N_4_ wt. %	MnO_2_ wt. %	Water Addition wt. %
1	88.5	6.0	2.0	1.5	2.0	14.0
2	86.0	7.0	2.5	1.5	3.0	14.0
3	83.5	8.0	3.0	1.5	4.0	14.0
4	81.0	9.0	3.5	1.5	5.0	14.0

**Table 3 micromachines-13-00550-t003:** Main functional parameters of the heat treatment process.

Functional Parameter	Variant 1	Variant 2	Variant 3	Variant 4
Dry raw material/glass foam amount (g)	470/462	470/463	470/462	470/464
Sintering/expanding temperature (°C)	920	927	934	942
Heating time (min)	34	36	38	41
Average rate (°C/min)				
heating	26.5	25.2	24.1	22.5
cooling	5.5	5.6	5.4	5.6
Index of growing the material volume	1.60	1.85	2.10	2.40
Specific consumption of energy (kWh/kg)	0.77	0.81	0.86	0.92

**Table 4 micromachines-13-00550-t004:** Main physical, thermal, mechanical and microstructural features of glass foam samples.

Feature	Variant 1	Variant 2	Variant 3	Variant 4
Apparent density (g/cm^3^)	0.49	0.38	0.32	0.28
Porosity (%)	76.7	81.9	84.8	86.6
Thermal conductivity (W/m·K)	0.110	0.089	0.080	0.071
Compressive strength (MPa)	4.3	3.9	3.5	2.8
Water absorption (vol. %)	3.3	3.3	3.4	3.1
Pore size (mm)	0.2–0.9	0.6–1.0	0.7–1.1	0.8–1.4

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
