# Peer review of "Glass Foam from Flat Glass Waste Produced by the Microwave Irradiation Technique"

_micromachines, 2022, doi:10.3390/mi13040550_

Round 1
Reviewer 1 Report
This is an interesting paper about a glass foam production experiment undertaken with the use of a home microwave. The authors present extensive analysis of the results and demonstrate good properties of the final material, such as thermal conductivity, and energy efficiency of the process. I recommend the paper for publication, with only a few corrections needed to introduce:
(1) page 3 / line 145: please look for the information about Hertz' experiments in Karlsruhe which are believed to be the first experimental demonstration of electromagnetic waves (and interestingly, their wavelength was exactly in the microwave range). This was in the 19th century, and not in the 1930s;
(2) page 7 / line 292: change "usually" to "usual" or "standard";
(3) page 10 / Fig. 3: due to low quality of the images, I would suggest to remove this figure. However, the authors could still provide information about the porosity determined from it.
Reviewer 2 Report
The article is devoted to the synthesis of foam glass from float glass using microwave irradiation.
The authors synthesized several compositions of foam glass with the introduction of various amounts of coal fly ash. As a result, foam glass samples were obtained with characteristics that allow them to be used as an insulating material.
There are a number of significant notes:
1) The first thing I would like to note is that the article does not correspond to the subject of this journal. From the “Aims&Scope”: “This journal seeks and encourages submissions on significant and original works related to all aspects of micro/nano-scaled structures, materials, devices, systems as well as related micro- and nanotechnology from fundamental research to applications”
2) The novelty of the work also raises questions. The authors write "The originality of the work was the application of the own technique of predominantly direct microwave heating, compared to the conventional heating methods commonly used in the manufacture of glass foam." , however, in the text we learn that work on the synthesis of foam glass using microwaves has already been carried out. The effect of coal fly ash has been studied in papers 11,12. What is the novelty of the work?
3) The authors write that “According to [9], previous experiments highlighted some disadvantages of using flat glass waste in the process of making glass foam. The Fe2O3 content of flat glass influenced the temperature and duration of the foaming process, causing the increase of these parameters…”. In container glass, Fe2O3 is also present, especially in colored glass. So I doubt that the matter is in Fe2O3. And the authors of the above work did not investigate the issue of the influence of iron oxide.
4) Is the fact of crystallization of glass positive or negative?
Of the less significant remarks:
1) 2nd link is not working
2) I would like to replace the photos in Figure 3 with better ones. At least 3b looks like a monitor screen was photographed.
Author Response
Please check the atachment

Reviewer 3 Report
This work reports the manufacture of glass foam from flat glass waste by using the advanced technique of predominantly direct microwave heating. The resultant glass foam has good thermal insulation characteristics (thermal conductivity of 0.089 W/m·K), high compressive strength (3.9 MPa) and a satisfactory microstructural homogeneity with pore size between 0.6-1.0 mm. Based on the advance of the homemade technique in producing good performance glass foams, I recommend that this manuscript should be accepted and published in Micromachines.
Round 2
Reviewer 2 Report
1) "Fundamental research and application" relates to micro- and nanotechnology. In the present work there are no elements of such technologies.
2) Authors claimed they novelty is simultaneous use of SiC and Si3N4. But as they wrote in introduction: "Experiments presented in [10] on the manufacture of glass foam from flat glass powder using simultaneously combinations of foaming agents such as carbides and nitrides (SiC, Si3N4 and AlN) with the addition of metal oxides with multiple valences (MnO2, Fe2O3 or CeO2) as oxygen-supplying agents in addition to the oxygen in the oven atmosphere.". So, this statement also can't be a novelty.
3) Moreover. Usually flat glass is used in buildings, that means that such glass muct be as much colorless as possible because of energy efficiency and aesthetic senses. So there is strong limits for Fe2O3 content in glass composition. And of course it is less that in container glass. So, this authors statement is incorrect.
4) Figure 4 is just an XRD plot. Some comments about influence of crystalline phases in foam glass on its properties must be adressed in manuscript.
Author Response
The aim of the current paper is the experimental small-scale manufacture of flat glass waste glass foam through technological changes in the current manufacturing process and the use of advanced technique of predominantly direct microwave heating, as well as the combined use of SiC and Si3N4 as foaming agents (unlike the association between Si3N4 and AlN tested in [10]) together with MnO2 as an oxygen-supplying agent. The combination of silicon carbide and nitride, adopted for their high microwave susceptibility, has not been tried, much less in microwave heating conditions. (pag. 4)
The formation of the crystalline phase of wollastonite in glass foam by using SiC as a foaming agent decreases the porosity and increases the mechanical strength of the foamed product. These effects influence to a small extent its physical and mechanical characteristics. (pag. 11)
- Wu, J.P.; Boccaccini, A.R.; Lee, P.D.; Kershaw, M.J.; Rawlings, R.D. Glass-ceramic foams from coal ash and waste glass: production and characterization. Adv. Appl. Ceram., 2006, 105(1), 32-39. (Referinte)
+ in Abstract (“microwave susceptible” si “first”)+ in Concluzii (“adopted for their microwave susceptibility”) + Pag.3 primul paragraph (“different”)
Contamination of flat glass batch from building demolition (which may contain spacer bars from sealed units, silicon carbide discs, floor sweepings, etc.) must be kept to a very low level. The industrial manufacturers of glass foam are not as interested of this type of glass waste compared to the container glass waste. Previous tests have shown that a significant increase of temperature and duration of the expansion process is possible due to some chemical components (including Fe2O3), which can influence the process parameters. Common methods of reducing these effects by using oxygen-supplying agents to improve the oxidation of the glass-based raw material can cause the macrostructural inhomogeneity of the foamed product. (pag. 2, paragraf 3)
Round 3
Reviewer 2 Report
This research doesn't apply to fundamental research on nano or microtechnology.
Despite some differences from previous papers, the results own poor scientific novelty and not high impact.